

# Comparative effectiveness of nonpharmacological interventions for the nutritional status of maintenance hemodialysis patients: a systematic review and network meta-analysis

Xiaolan Ma[1,*], Chun Tang[2,*], Hong Tan[1], Jingmei Lei[1] and Li Li[3,4]

[1] School of Nursing, Xinjiang Medical University, Urumqi, Xingjiang, China
[2] Neonatal Intensive Care Unit, the First Affiliated Hospital of Xinjiang Medical University, Urumqi, Xinjiang, China
[3] Department of Urology, the First Affiliated Hospital of Xinjiang Medical University, Urumqi, Xinjiang, China
[4] Health Care Research Center for Xinjiang Regional population, Urumqi, Xinjiang, China
* These authors contributed equally to this work.

## ABSTRACT

**Objective:** We aim to analyze the effects of different nonpharmacological interventions on the nutritional status of patients on maintenance hemodialysis (MHD).

**Methods:** Randomized controlled trials (RCTs) conducted up to October 12, 2023 were searched in the Cochrane Library, Chinese National Knowledge Infrastructure, Wan Fang Database, VIP databases, and China Biomedical Literature Database. R and Review Manager software were used for data analysis, the quality of the literature was assessed using the Cochrane Risk Bias Tool RoB2.0, the reliability of evidence was evaluated using Grading of Recommendations, Assessment, Development, and Evaluation guidelines, and sources of heterogeneity were explored through sensitivity analyses. This study was registered in PROSPERO with registration number CRD42023458187.

**Results:** A total of 54 studies met the criteria, 3,861 patients were enrolled in this study, and 11 interventions were explored. The results of the network meta-analysis (NMA) showed that dietary intervention is the best intervention in terms of improving patients' body mass index and serum albumin levels and health education is the best intervention in terms of improving patients' hemoglobin levels.

**Conclusion:** This NMA confirmed that different nonpharmacological interventions benefit the nutritional status of patients on MHD, providing novel insights for healthcare practitioners. However, high-quality RCTs should be designed to validate the stability of the level of evidence for different nonpharmacological interventions.

Corresponding author
Li Li, 1311016216@qq.com

## INTRODUCTION

Maintenance hemodialysis (MHD) is one of the essential treatment modalities (*Liyanage et al., 2015*). National Hemodialysis Case Information Registration System (Chinese National Renal Data System) statistics showed that the number of hemodialysis (HD) patients in China was approximately 749,600 at the end of 2021, which was 3.2-fold that in 2011. Thus, China has the highest number of MHD patients in the world (*Fan et al., 2023*). MHD is often associated with a wide range of complications, including malnutrition, which has a prevalence of more than 60% (*Ruperto & Barril, 2023*), one of the most common chronic complications of MHD (*Wu, Wang & Zhou, 2019*), and one of the most important factors contributing to patients' complications and clinical outcomes (*Miu, Wang & Bao, 2022*; *Sahathevan et al., 2020*; *Viramontes-Hörner et al., 2022*). Effective nutritional interventions can improve clinical outcomes related to mortality and morbidity in patients on MHD and increase patient survival (*Limwannata et al., 2021*). Therefore, effective interventions and treatment programs may prevent the occurrence of malnutrition in patients on MHD and improve their nutritional indicators.

Interventions for the nutritional status of MHD patients include pharmacological interventions, including treatments with levocarnitine, growth hormones, and anabolic steroids, and nonpharmacological interventions. Unfortunately, pharmacological treatments have many adverse reactions, such as nausea, vomiting, and dizziness, and increase the risk of cardiovascular diseases. Thus, the safety and efficacy of these interventions need to be verified (*Lu et al., 2021*; *Oliveira et al., 2019*). Nonpharmacological interventions do not use medications designed to improve a patient's quality of life or healthy lifestyle (*Fu et al., 2020*), including exercise therapy, dietary therapy, health education, oral nutritional supplements (ONS), and combined therapies, which can effectively improve patients' nutrition-related indicators. These treatments can reduce various complications, thus improving quality of life (*Abreu et al., 2017*; *Jeong et al., 2019*; *Martin-Alemañy et al., 2022*; *Shi et al., 2021*; *Sun, Sun & Yang, 2022a*). However, no standardized criteria for determining what intervention should be administered have been established, and whether nonpharmacological interventions are more effective than pharmacological interventions in improving nutritional indicators is unclear. Each nonpharmacological intervention has unique strengths and potential mechanisms of intervention, varying in therapeutic efficacy in improving nutritional status. For example, exercise can lead to altered body composition (*Abreu et al., 2017*), and nutritional supplements can directly affect biomarker levels (*e.g.*, protein and iron; *Yang et al., 2021*). Biochemical indicators, such as hemoglobin (HB) and serum albumin (ALB), and anthropometric indicators, such as body mass index (BMI), are often used in measuring the recent nutritional status of MHD patients, and most researchers use the above indicators to measure the effectiveness of a nonpharmacological intervention on relevant nutritional indicators; these indicators reflect improvements in the nutritional status of MHD patients (*Abreu et al., 2017*; *Jeong et al., 2019*; *Martin-Alemañy et al., 2022*; *Shi et al., 2021*; *Sun, Sun & Yang, 2022a*), and HB is an important indicator that reflects the quality of long-term survival and prognoses of patients (*Li et al., 2022*). ALB is one of the main

biochemical indicators for the nutritional assessment of dialysis patients (*Chen, 2020*), and BMI is an important indicator of the prognoses of dialysis patients (*Suzuki et al., 2020*). Therefore, the above indicators were selected as outcome indicators in the measurement of change in nutritional status of MHD patients in this study.

Traditional meta-analysis and related studies have shown that nonpharmacological interventions can improve nutritional status and body composition (*Bakaloudi et al., 2020*; *Huang, Huang & Xing, 2015*; *Liu et al., 2023a*; *Mah et al., 2020*; *Wu et al., 2022*) but are limited to ONS, exercise training, and analysis of three interventions for patients taking ONS and undergoing exercise training. Moreover, these studies only compared two interventions. Network meta-analysis (NMA) refers to the use of the superiority or inferiority ranking of the effect of each intervention on the improvement of relevant outcome indicators, ranking various interventions and identifying the best interventions (*Lu & Ades, 2004*). Therefore, in this study, we conducted an NMA to compare the effects of different nonpharmacological interventions on the nutritional status of patients on MHD and search for the most reliable current nonpharmacological interventions to improve nutritional indicators. The aim was to provide a strong basis for selecting appropriate interventions for the adoption of optimal and effective methods for improving the nutritional status of patients.

## METHODS

This systematic review strictly followed COCHRANE's Preferred Reporting Items for Systematic Reviews and Meta-Analyses (PRISMA) guidelines (*Page et al., 2021*) and the "Prioritized Reporting of Network Meta-Analyses Entry: an Interpretation of the PRISMA Extension Statement" (PRISMA Extension Statement for Reporting of Systematic Reviews Incorporating Network Meta-analyses; (*Hutton, Catalá-López & Moher, 2016*). The NMA is registered on PROSPERO under registration number CRD42023458187.

### Data sources and searches

The literature was searched independently by two investigators according to the search formula, and any disagreement about the included studies was discussed and resolved by a third investigator and by searching PubMed, Web of Science, Embase, Cochrane Library, Chinese National Knowledge Infrastructure, Wan Fang Database, China Science and Technology Journal Database (VIP Database), and China Biomedical Literature Database. The search was conducted from the inception of a database to October 12, 2023. Searches were conducted using a combination of subject terms: for English, "Hemodialysis/Renal dialysis/MHD/Dialysis/Hemoperfusion"; "nutritional status/malnutrition"; "Exercise/ Psychology/Diet/Health Education/Oral nutritional supplements," and for Chinese writing, "Hemodialysis/MHD/renal dialysis/dialysis/end-stage renal disease"; "Nutrition/ nutritional status/malnutrition/nutritional deficiencies"; "Exercise/Psychology/Diet/ Health Education/Oral Nutritional Supplementation" (Appendix A).

## Inclusion and exclusion criteria

We developed inclusion criteria according to the PICOS principles, including study population, intervention, control, outcome indicators, and study design.

### Population

We included MHD patients who met the diagnostic criteria for CKD or ESRD and who were eligible to receive dialysis for ≥3 months, were ≥18 years of age, and were on dialysis ≥2 times per week.

### Interventions

Nonpharmacological interventions based on routine dialysis, including dietary intervention, exercise therapy (*e.g.*, aerobic exercise, resistance exercise, and aerobic exercise combined with resistance exercise), ONS, health education, combination therapies (*e.g.*, ONS combined with resistance exercise, ONS combined with aerobic exercise, and ONS combined with aerobic and resistance exercises) and comprehensive nursing care were included. The definitions of the interventions are detailed in Table 1.

### Comparisons

Usual care or nonpharmacological interventions different from those of the intervention group were used.

### Outcomes

The primary outcome indicators in this study were HB, ALB, and BMI, which were measured using mean difference (MD) ± standard deviation (SD) because the outcome indicators were continuous variables.

### Study design

Randomized controlled trials (RCTs) that reported interventions and corresponding intervention outcomes were included.

## Literature exclusion criteria

We excluded (1) duplicate publications; (2) full text of the study was not available; (3) literature that was not in Chinese or English; (4) literature with incomplete data and unclear outcome effects.

## Data selection and extraction

Study selection: Three researchers (Xiaolan Ma, Hong Tan, and Jingmei Lei) independently screened the literature back-to-back strictly according to the inclusion and exclusion criteria and eliminated duplicates, systematic evaluations, and reviews by using NoteExpress literature management software. By reading the titles and abstracts and then read the full texts, they further excluded studies that did not match the content of the present study. Inconsistencies encountered were discussed by two researchers (Li Li and Chun Tang) and decided whether to include a study.

**Table 1 The definitions of the interventions.**

| | |
|---|---|
| Dietary intervention | According to the nutritional needs, health status, food preferences and other factors of individuals or groups, by adjusting dietary structure, food intake and eating habits, the main causes of nutritional disorders and food categories were found, so as to formulate effective nursing measures to prevent and improve the nutritional status of hemodialysis patients (*Vergili et al., 2024*). |
| Health education | Through planned, organized and systematic social and educational activities covering nutrition, mental health, disease prevention and other key areas, the aim is to guide people to take the initiative to adopt healthy behaviors and lifestyles, reduce health risks, prevent diseases, and improve the quality of life. It is delivered in a variety of ways, including lectures, seminars, online courses and educational materials (*Xu, Xu & Zhou, 2023*). |
| ONS | It is a form of nutritional supplementation made from protein, energy or a combination thereof, including macronutrients (proteins, carbohydrates, fats) and micronutrients (vitamins, minerals, trace elements) (*Jensen, 2013*). |
| Resistance exercise | Refers to a form of exercise in which a certain amount of resistance is applied to promote muscle growth and strength gain, which may be provided by another person, one's own limbs, or equipment (*e.g.*, sandbags, elastic bands, *etc.*) (*Faigenbaum & Myer, 2010*) |
| Aerobic exercise | Any continuous traditional aerobic exercise pattern (*e.g.*, walking, running, cycling, rowing, swimming, circuit training, stepping exercises) (*Voet et al., 2019*) |
| Comprehensive nursing care | It includes the combination of dietary intervention, health education, oral nutritional supplements, aerobic exercise, resistance exercise, and other two or more intervention methods, among which aerobic and resistance exercise were collectively referred to as exercise training. |

Data extraction: Data were organized using Excel 2021. The data were extracted by Xiaolan Ma, Hong Tan, and Jingmei Lei, including (1) the basic information of the included studies, such as the first author, year of publication, and country; (2) baseline characteristics of the study subjects, interventions, and control measures; and (3) corresponding outcome indicators of the included studies. Any dispute during data extraction was resolved by two researchers (Li Li and Chun Tang).

## Literature quality assessment
### Risk of bias assessment

Three researchers (Xiaolan Ma, Hong Tan, and Jingmei Lei) independently assessed the quality of the included research literature using the Cochrane Risk of Bias Assessment Tool RoB2.0 (*Sterne et al., 2019*). The likelihood of bias in the RCTs was evaluated according to the following five domains: (1) bias in the randomization process; (2) bias in deviation from established interventions; (3) bias in missing outcome data, (4) bias in outcome measurement; and (5) bias in selective reporting of outcomes. The risk of bias for each domain was categorized as low risk of bias, some concerns, and high risk. The overall risk of biased judgment represented the quality of the literature. For each selected article, the risk of bias was assessed independently by two researchers, and when no agreement was reached, another researcher (Li Li) was consulted. When one of the five domains was rated as "high risk," the overall bias was rated as "high risk"; when one of the five domains was rated as "some concerns," the overall bias was rated "some concerns"; when all five domains were rated as "low risk," the overall bias was rated as "low risk." The funnel plot was drawn with the "netmeta" package in R 4.2.3 software, and the presence of publication bias was assessed with Egger's test.

### Quality of evidence

According to the Grading of Recommendations, Assessment, Development, and Evaluation (GRADE) guidelines, the level of evidence is categorized into four levels: high, moderate, low, and very low. Directly comparable evidence is initially high-quality evidence, which is assessed in terms of five dimensions, namely, risk of bias, inconsistency, imprecision, indirectness, and publication bias. If a degradation is present in one of the five items, then the evidence has a medium level; if two are degraded, then the evidence has a low level; and if three or more are degraded, the evidence has a very low level (*Balshem et al., 2011*; *Puhan et al., 2014*). Network evidence was finally considered low because of inconsistency or imprecision (*Puhan et al., 2014*), and we rated imprecision according to the GRADE guidelines (*Zeng et al., 2021*). When the 95% credibility interval (credibility interval, CrI) exceeded more than one significance threshold, we lowered the imprecision by two levels (*Zeng et al., 2022*). Three researchers (Xiaolan Ma, Hong Tan and Jingmei Lei) independently rated the included literature, and any discrepancy among the three researchers were resolved by consulting two researchers (Li Li and Chun Tang). The results of the final ratings were presented in an NMA summary of findings (*Brignardello-Petersen et al., 2021*; *Wang et al., 2020*).

## Statistical analysis

R 4.2.3 software was used to run the Gemtc package under the R studio environment. We used Bayesian for NMA and four Monte Carlo Markov chains for each model to set initial values. The number of iterations was 50,000, and a burn-in 20,000 iterations were used for eliminating the effects of the initial values. The convergence of the models was evaluated using the potential scale reduced factor (PSRF), and the closer the PSRF value was to 1, the better the convergence of iterations was (*Brooks & Gelman, 1998*). According to the deviance information criterion (DIC) value, a random- or fixed-effects model was selected for analysis. A difference of ≤5 between the two models indicated that the two model fits are consistent, and the model with a small $I^2$ value was selected. When the difference between the two models had a DIC value of >5, the model with a small DIC value was selected for analysis (*TianJingHui, 2017*). Inconsistency was assessed using inconsistency factor and node-splitting method. When the 95% CrI of the inconsistency factor contains 0 in the node-splitting method and $P > 0.05$, no statistical inconsistency is present, an indirect comparison is consistent with the results of a direct comparison, and analysis is conducted using a consistency model; otherwise, the inconsistency model is used (*White et al., 2012*). In addition, the efficacy of different interventions was ranked using the surface under the cumulative ranking (SUCRA), which ranges from 0 to 100%, where $0 < $ SUCRA $\leq 100\%$; 100% indicates effective intervention and 0 indicates poor and ineffective. Funnel plots were created using the "netmeta" package in R 4.2.3 software, and the presence of publication bias was assessed with Egger's test and the sources of heterogeneity were explored using sensitivity analyses.

# RESULTS

## Literature search results

A total of 14,309 documents were initially searched, of which five were obtained by tracing the references of the examined documents, and 13,348 duplicate studies and irrelevant titles and abstracts were excluded. The remaining 961 studies were viewed in their entirety, and 54 were included in the systematic evaluation and NMA. The specific screening process is shown in Fig. 1.

## Characteristics of the included studies

A total of 54 studies were included in this study, encompassing 3,861 patients and 11 interventions. Three of these studies were three-armed studies (*Afshar et al., 2010*; *Jeong et al., 2019*; *Martin-Alemaňy et al., 2020*), and the remaining studies were two-arm studies. The information and baseline characteristics of the studies are shown in Table 2.

## Risk of bias of the included studies

A total of 54 studies were included in this study. The risk of bias for each trial was assessed using the Cochrane Risk of Bias Assessment Tool RoB2.0: 13 studies were assessed to have a low risk of bias, 29 studies were considered "some concerns," and 12 studies were assessed to have a high risk of bias. Of the 54 studies included, only two studies had incomplete data results because the difference in the number of lost visits between the control and experimental groups was significant. Thus, they were rated as high risk (Figs. 2 and 3).

## Results of NMA

### Network plot

The network evidence maps for each outcome metric are shown in Fig. 4. Nine nonpharmacological interventions were analyzed for their effects on BMI values in patients on MHD; 11 nonpharmacological interventions were analyzed for their effects on ALB levels in patients on MHD; and 10 nonpharmacological interventions were analyzed for their effect on HB levels in patients on MHD.

### Results of NMA on the effects of different interventions on BMI in patients on MHD

A total of 28 studies (*Abreu et al., 2017*; *Ai, 2020*; *Bolasco et al., 2011*; *Cai et al., 2022*; *Chen, Zhao & Huang, 2019*; *Dai & Ma, 2021*; *Dong et al., 2011*; *Hristea et al., 2016*; *Jeong et al., 2019*; *Kozlowska et al., 2023*; *Leng, 2012*; *Li & Feng, 2020*; *Liao et al., 2016*; *Limwannata et al., 2021*; *Lu, 2022*; *Martin-Alemaňy et al., 2020*; *Sezer et al., 2014*; *Tan et al., 2015*; *Tayebi, Ramezani & Kashef, 2018*; *Vijaya et al., 2019*; *Wang, 2018*; *Wang, 2019*; *Wang et al., 2023*; *Wen et al., 2022*; *Wilund et al., 2010*; *Xu & Fang, 2016*; *Yang et al., 2021*; *Yu & Cao, 2018*) assessed changes in BMI, and dietary intervention (MD = 1.81, 95% CrI [0.49–3.06]; $P < 0.05$) was superior to usual care in improving BMI. None of the other interventions and none of the comparisons showed statistical significance ($P > 0.05$; Table 3).

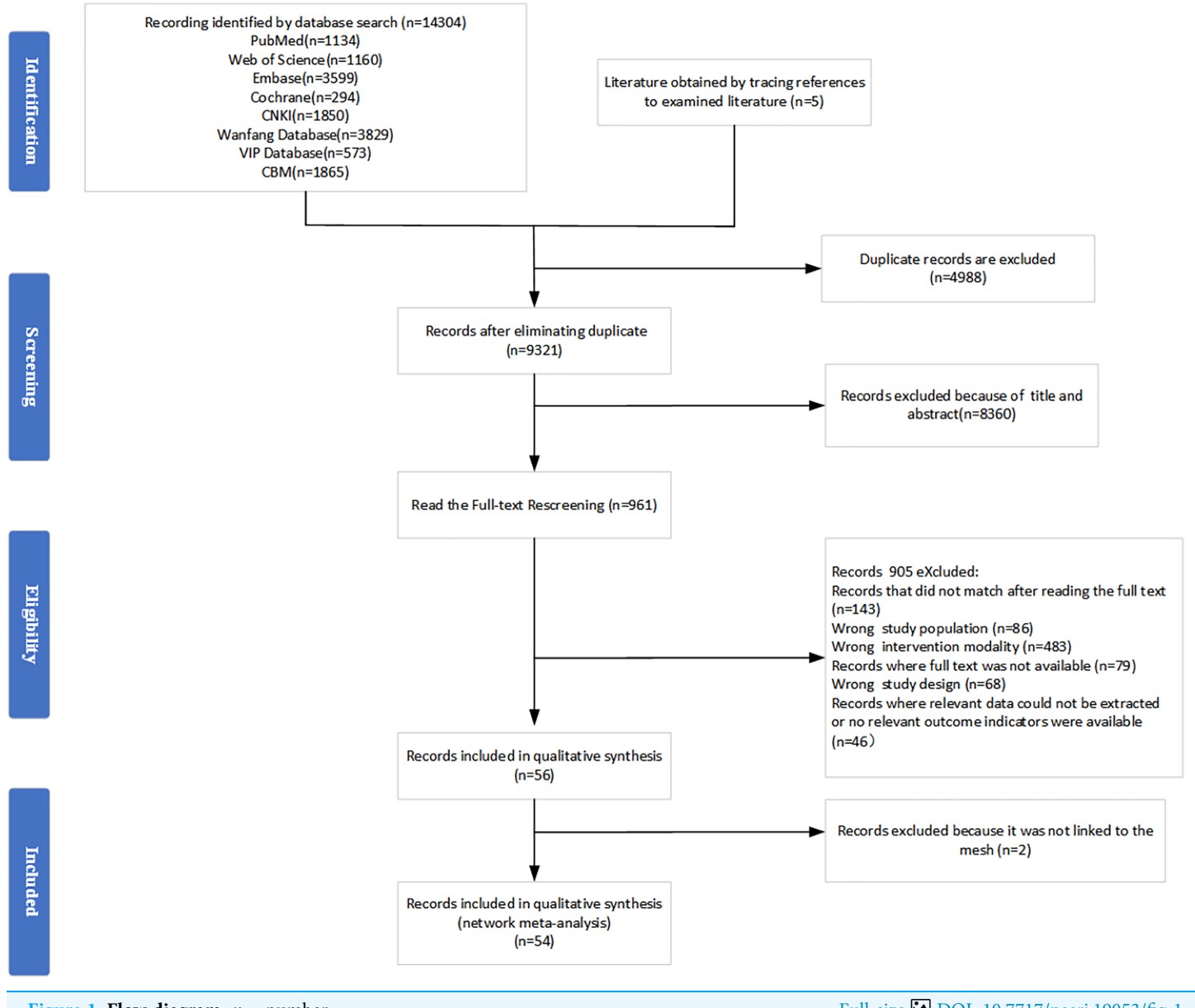

**Figure 1 Flow diagram.** *n* = number.

### Results of NMA on the effects of different interventions on ALB in patients on MHD

A total of 47 studies (*Abreu et al., 2017*; *Afshar et al., 2010*; *Ai, 2020*; *Bolasco et al., 2011*; *Cai et al., 2022*; *Chen, Zhao & Huang, 2019*; *Dai & Ma, 2021*; *Deng, 2011*; *Feng et al., 2020*; *Frih et al., 2017*; *Hristea et al., 2016*; *Jeong et al., 2019*; *Kozlowska et al., 2023*; *Leng, 2012*; *Li & Feng, 2020*; *Li et al., 2008*; *Liao et al., 2016*; *Limwannata et al., 2021*; *Lu, 2022*; *Martin-Alemañy et al., 2020*; *Martin-Alemañy et al., 2022*; *Martin-Alemañy et al., 2016*; *Sezer et al., 2014*; *Shi et al., 2021*; *Su et al., 2022*; *Sun, Sun & Yang, 2022a*; *Tabibi et al., 2023*; *Tan et al., 2015*; *Tayebi, Ramezani & Kashef, 2018*; *Wang & Liu, 2021*; *Wang, 2018*; *Wang et al., 2019*; *Wang, 2019*; *Wang et al., 2023*; *Wei, 2020*; *Wen et al., 2022*; *Wilund et al., 2010*; *Xu et al.,*

**Table 2 Basic characteristics of the included studies.**

| Study (year) | Country | Sample I | Sample C | Type of intervention I | Type of intervention C | Intervention duration | Outcomes |
|---|---|---|---|---|---|---|---|
| *Wei (2020)* | China | 42 | 42 | Comprehensive nursing care: Exercise training + health education | Usual care | – | ①② |
| *Zhou et al. (2016)* | China | 30 | 30 | Comprehensive nursing care: Exercise training + health education + dietary intervention | Usual care | 24 weeks | ①② |
| *Li et al. (2008)* | China | 23 | 25 | Comprehensive nursing care: dietary intervention + health education | Usual care | 24 weeks | ①② |
| *Lu (2022)* | China | 30 | 28 | Comprehensive nursing care: Exercise training + health education + dietary intervention | Usual care | 24 weeks | ①②③ |
| *Abreu et al. (2017)* | Brazil | 25 | 19 | Resistance exercise: Applying ankle-cuffs and elastic bands for training | Usual care | 12 weeks | ①②③ |
| *Wang et al. (2019)* | China | 20 | 20 | Aerobic exercise: Cycling exercise program | Usual care | 24 weeks | ①② |
| *Afshar et al. (2010)* | Iran | 7 | 7 | Resistance exercise: Applying ankle weights for knee extension flexion and hip abduction-flexion for training | Usual care | 8 weeks | ①② |
| *Dai & Ma (2021)* | China | 20 | 30 | Resistance exercise | Usual care | 24 weeks | ①②③ |
| *Yan, Zhao & Peng (2022)* | China | 47 | 47 | Resistance exercise: Applying exercise equipment for training, such as sandbags | Usual care | 12 weeks | ①② |
| *Cai et al. (2022)* | China | 44 | 44 | Resistance exercise: Applying sandbags, elastic bands, *etc.* for training | Usual care | 24 weeks | ①②③ |
| *Tayebi, Ramezani & Kashef (2018)* | Iran | 17 | 14 | Resistance exercise: Included the isometric grip exercise and an isometric contraction of the leg and core | Usual care | 8 weeks | ②③ |
| *Liao et al. (2016)* | China | 20 | 20 | Aerobic exercise: Cycling exercise program | Usual care | 12 weeks | ②③ |
| *Leng (2012)* | China | 39 | 42 | Aerobic exercise: Walking, brisk walking, jogging, tai chi, *etc.* | Usual care | 24 weeks | ①②③ |
| *Wang & Liu (2021)* | China | 26 | 28 | Aerobic exercise: Cycling exercise program | Usual care | 24 weeks | ①② |
| *Afshar et al. (2010)* | Iran | 7 | 7 | Aerobic exercise: Cycling exercise program | Usual care | 8 weeks | ①② |
| *Wilund et al. (2010)* | US | 8 | 9 | Aerobic exercise: Cycling exercise program | Usual care | 16 weeks | ②③ |
| *Su et al. (2022)* | China | 45 | 45 | Aerobic exercise: Cycling exercise program, brisk walking | Usual care | – | ①② |
| *Wang et al. (2023)* | China | 43 | 41 | ONS: A variety of substances such as protein, fat, carbohydrates, water, multivitamins, minerals and whey | Usual care | 12 weeks | ①②③ |
| *Chen, Zhao & Huang (2019)* | China | 54 | 54 | ONS: Including protein, glutamine component, energy and iron, calcium elements | Usual care | 24 weeks | ①②③ |
| *Jeong et al. (2019)* | US | 45 | 44 | ONS:30 grams of whey | Usual care | 48 weeks | ②③ |
| *Limwannata et al. (2021)* | Thailand | 30 | 24 | ONS: Including protein, Carbohydrate and Fat | Usual care | 4 weeks | ②③ |
| *Bolasco et al. (2011)* | Italy | 15 | 14 | ONS: Oral amino acid supplementation | Usual care | 12weeks | ①②③ |

(Continued)

| Study (year) | Country | Sample I | Sample C | Type of intervention I | C | Intervention duration | Outcomes |
|---|---|---|---|---|---|---|---|
| *Tan et al. (2015)* | China | 31 | 31 | ONS: Including energy, protein, fat and carbohydrate | Usual care | 12 weeks | ①②③ |
| *Yang et al. (2021)* | China | 120 | 120 | ONS: Including energy, fat and unsaturated fatty acids et al | Usual care | 12 weeks | ①②③ |
| *Wen et al. (2022)* | China | 49 | 43 | ONS: Oral nonprotein calorie jelly: including energy, fat and carbohydrate | Usual care | 24 weeks | ①②③ |
| *Sezer et al. (2014)* | Turkey | 29 | 29 | ONS: including protein, energy, fat, carbohydrate, sodium, potassium and phosphorus | Usual care | 24 weeks | ①②③ |
| *Yu & Cao (2018)* | China | 31 | 31 | ONS: Doesn't state nutritional content | Usual care | 12 weeks | ②③ |
| *Ai (2020)* | China | 40 | 40 | Dietary intervention: Develop an individual programme to provide 1.2–1.4 g/kg of high-quality protein (eggs, milk, lean meat) and low-fat diet; adjust sugar, potassium and sodium intake with nutritional indicators; supplement with micronutrients; and provide guidance on dietary measurements and record-keeping | Usual care | 12 weeks | ①②③ |
| *Sun, Sun & Yang (2022a)* | China | 60 | 59 | Dietary intervention: According to the 2021 edition of the Chinese Nutrition Guidelines for Chronic Kidney Disease (*Chinese Society of Nephrologists CMDA, Expert Collaboration Group of Nutritional Therapy Guidelines of Kidney Disease Committee of Chinese Society of Integrated Traditional Chinese and Western Medicine, 2021*), the diet plan was formulated: 1) accurate measurement, distribution of scale instruments and quantitative control table; 2) accurately record food types and portions using diary cards; 3) accurate assessment and adjustment of diet plan according to biochemical test results; 4) Accurate correction of deviations to correct dietary misunderstandings and defects of patients | Usual care | 24 weeks | ①② |
| *Zhou (2020)* | China | 37 | 37 | Dietary intervention: Individualized dietary regimen according to the patient's condition. High-quality protein (milk, chicken, lean meat, *etc.*), alpha keto acid tablets if necessary, low-fat diet (1.5 g/(kg/d), sugar 5 g/(kg/d), vitamin and trace element supplementation, and enhanced education | Usual care | – | ①② |
| *Xu et al. (2022)* | China | 25 | 25 | Dietary intervention: After the evaluation of patients, a personalized diet plan was formulated, containing high-quality protein (eggs, fish, lean meat, *etc.*) accounting for more than 50% of the intake of food, sugar 5–6 g/kg, sodium 6 g/d, plant fat, vitamins and trace elements, and strengthening health education | Usual care | – | ①② |
| *Wang (2019)* | China | 37 | 37 | Dietary intervention: Nutritionist customized program, fat 1.5g/kg, sugar 5g/kg, high-quality protein (eggs, milk, meat), α-keto acid tablets, vitamin and trace elements supplement, strengthen health education | Usual care | 24 weeks | ①②③ |
| *Wang (2018)* | China | 30 | 30 | Dietary intervention: Nutritionist customized personalized diet plan. Fat 1.5 g/kg, sugar 5 g/kg, high quality protein (eggs, milk, lean meat) 1.2 gkg, can add α-keto acid, vitamin and trace elements, strengthen health education. | Usual care | – | ①②③ |
| *Deng (2011)* | China | 48 | 48 | Dietary intervention: The patients' nutritional status and dietary habits were evaluated to develop individualized diet plans. Contain high quality protein (eggs, fish, lean meat, *etc.*), calories, fat and other nutrients supplement; Water and electrolyte regulation included potassium, sodium salt, phosphorus (0.6–1.2 g/d); Vitamins and trace elements were supplemented, and health education was strengthened | Usual care | 96 weeks | ①② |

| Study (year) | Country | Sample I | Sample C | Type of intervention I | C | Intervention duration | Outcomes |
|---|---|---|---|---|---|---|---|
| *Li & Feng (2020)* | China | 20 | 20 | Dietary intervention: According to the patient's condition, the diet plan was customized: 200–250 g grain per day, fat accounting for 20–30% of total energy, glucose <50 g (≤10% of total energy), protein 1.5 g/kg (preferably chicken, fish and beef), and calorie 146 kcal/kg. Instruction in proper cooking techniques, such as potassium removal of vegetables, and adjustment of diet based on biochemical examinations | Usual care | – | ②③ |
| *Xu & Fang (2016)* | China | 46 | 46 | Dietary intervention: According to the patient's condition, the Nutritionist made a personalized diet plan: fat 15 g/kg, sugar 5 g/kg, supplemented lean meat, milk, eggs and other high-quality protein, oral α-keto acid tablets, vitamin and trace elements, and strengthened health education | Usual care | 24 weeks | ①②③ |
| *Kozlowska et al. (2023)* | Poland | 35 | 35 | Dietary intervention: Individualized diet plans according to European guidelines (*Fouque et al., 2007*), meals provided by diet catering companies with nutritional values of about 20%± of the average requirement: energy 416 ± 23 kcal, protein 16.4 ± 0.6 g, fat 14.2 ± 1.6 g, total carbohydrates 58.2 ± 5.6, potassium 452.7 ± 79.6 mg, and phosphorus 183.9 ± 10.5 mg | Usual care | 24 weeks | ①②③ |
| *Feng et al. (2020)* | China | 68 | 68 | Health education: Electronic Intelligent Tool + Nutritional Guidance: (1) build a WeChat group to guide patients and their families to use the personalized dietary application for kidney disease; (2) enter patient information into the WeChat public number applet to automatically generate a daily diet plan; (3) the applet recommends recipes, provides illustrations of the amount of food, and the patient chooses his/her own food; and (4) the patient eats according to the recommendation and provides feedback on the use of the applet on a monthly basis | Usual care | 48 weeks | ①② |
| *Zeng et al. (2020)* | China | 50 | 50 | Health education: (1) Establish wechat group and QQ group, including supervisor nurses, dietitians, specialists and patients; (2) nurse-in-charge provided psychological guidance, enhanced diet implementation, and encouraged sports and sports activities; (3) Network assistance of dietitian for results feedback, answers to nutrition knowledge, recipe adjustment, and personalized guidance; (4) The whole process is supervised by specialists | Usual care | 24 weeks | ①② |
| *Shi et al. (2021)* | China | 79 | 78 | Health education: Hemodialysis specialist nurses, nephrologists, nutrition managers and medical information technicians. The head nurse was responsible for the overall arrangement, organization of training and content writing, and the team solved the problems in the intervention together. (2) Nutrition education module: Information technology engineers set up a "nutrition learning garden for hemodialysis patients" on the hospital platform, which was exclusive to the experimental group. Researchers designed individualized nutrition programs and pushed them through columns. Patients can log in with ID and leave messages for interaction. Researchers replied in time, and designed the next push content according to the feedback. Content is not repeated, and past content remains searchable. The team designed the push content according to the evaluation results and literature | Usual care | 12 weeks | ①② |
| *Vijaya et al. (2019)* | India | 139 | 138 | Health education: Dietary patterns are developed by a renal dietitian on a monthly basis based on the patient's condition. Dietary guidance is also provided to dialysis patients in the form of one-on-one counselling or group counselling, and hands-on nutritional education sessions are provided to family members and friends of the patients | Usual care | 24 weeks | ③ |

(Continued)

| Table 2 (continued) | | | | | | | |
|---|---|---|---|---|---|---|---|
| Study (year) | Country | Sample | | Type of intervention | | Intervention duration | Outcomes |
| | | I | C | I | C | | |
| *Yao et al. (2020)* | China | 48 | 46 | Health education: (1) Establish a diet education group: learn the Diet Education Manual for maintenance hemodialysis Patients, diet management, wechat and peer support, forgetting curve and other knowledge. (2) Evaluation: patients' diet management behavior was evaluated and graded into excellent, good and poor levels, individualized education programs were formulated, and patients were taught to use Internet and wechat to receive education. (3) Diet education: hierarchy: nurses were assigned according to the score of diet management behavior; Stage type: according to the forgetting curve, they were divided into early stage, middle stage and late stage, and different intensities of education were given. Diversification: distribution of health education manuals, combined with individual health education, according to the score of diet management behavior was divided into three grades: excellent, medium and poor. A 3-day food diary card was used to record food intake. Wechat group, video lectures, peer support, *etc.* were selected according to the patient's condition. (4) Follow-up management: outpatient, wechat, telephone and home visits, and follow-up frequency was adjusted according to the situation | Usual care | 24 weeks | ①② |
| *Fakhrpour et al. (2020)* | Iran | 24 | 21 | Aerobic and Resistance exercise: Cycling exercise program + Resistance exercise of the lower | Usual care | 16 weeks | ① |
| *Tabibi et al. (2023)* | Iran | 35 | 33 | Aerobic and Resistance exercise: Aerobic exercise included moving legs back and forth, shoulder abduction and adduction et al; Resistance exercise included exercises for the upper and lower limbs as well as core strength exercises using body weight, weight cuffs, dumbbells, and elastic bands of varying intensity | Usual care | 24 weeks | ①② |
| *Fang et al. (2023)* | China | 29 | 30 | Aerobic and Resistance exercise: Cycling exercise + Resistance exercise for the upper and lower limbs | Usual care | 12 weeks | ① |
| *Frih et al. (2017)* | Tunisia | 21 | 20 | Aerobic and Resistance exercise: Aerobic exercise included endurance program included ergometer cycling and treadmill walking; Resistance exercise included dynamic closed and open-chain strengthening exercises. Such as the quadriceps muscles, pectoral muscles, triceps brachia muscles, biceps brachia muscles, and hamstrings were trained on a multigym | Usual care | 16 weeks | ①② |
| *Zhu et al. (2020)* | China | 53 | 53 | Aerobic and Resistance exercise: Aerobic exercise included cycling exercise, jogging ea al; Resistance exercise inculded sandbags, stretch cords, et al | Usual care | 12 weeks | ①② |
| *Martin-Alemañy et al. (2020)* | Mexico | 12 | 13 | ONS combined with Aerobic exercise: ONS included energy, Protein, Carbohydrate and Fat et al.; Aerobic exercise: Pedaling astationary bike | ONS | 12 weeks | ①②③ |
| *Martin-Alemañy et al. (2016)* | Mexico | 17 | 19 | ONS combined with Resistance exercise: ONS included energy, Protein, Fat and vitamins et al; Resistance exercise: Patients were trained according to an adaptation of the program "Exercise: A Guide for People on Dialysis" (*Painter, 2000*) | ONS | 12 weeks | ② |
| *Martin-Alemañy et al. (2020)* | Mexico | 9 | 13 | ONS combined with Resistance exercise: ONS included energy, Protein, Carbohydrate and Fat et al; Resistance exercise: Patients were trained according to an adaptation of the program "Exercise: A Guide for People on Dialysis" (*Painter, 2000*) | ONS | 12 weeks | ①②③ |
| *Jeong et al. (2019)* | US | 49 | 44 | ONS combined with Aerobic exercise: oral protein supplement (30 grams of whey) ; Aerobic exercise: cycle ergometers | Usual care | 48 weeks | ②③ |

| Study (year) | Country | Sample I | C | Type of intervention I | C | Intervention duration | Outcomes |
|---|---|---|---|---|---|---|---|
| *Hristea et al. (2016)* | France | 7 | 9 | ONS combined with Aerobic exercise: ONS: adjusted to EBP guideline settings and patient needs (*Magnard et al., 2013*); Aerobic exercise: submaximal individualized cycling exercise using a cycloergometer (*Magnard et al., 2013*) | ONS | 24 weeks | ①②③ |
| *Martin-Alemañy et al. (2022)* | Mexico | 10 | 14 | ONS combined with Aerobic and Resistance exercise: ONS included energy, Protein, Carbohydrate and Fat et al; Aerobic exercise: Cycling exercise program; Resistance exercise: Patients were trained according to an adaptation of the program "Exercise: A Guide for People on Dialysis" (*Painter, 2000*) | ONS | 24 weeks | ② |
| *Dong et al. (2011)* | US | 10 | 12 | ONS combined with Resistance exercise: ONS included energy, Protein, Carbohydrate and Fat et al; Resistance exercise: A pneumatic leg press machine | ONS | 24 weeks | ①③ |

**Note:**
①:HB; ②:ALB; ③:BMI. I = Intervention group; C = Control group

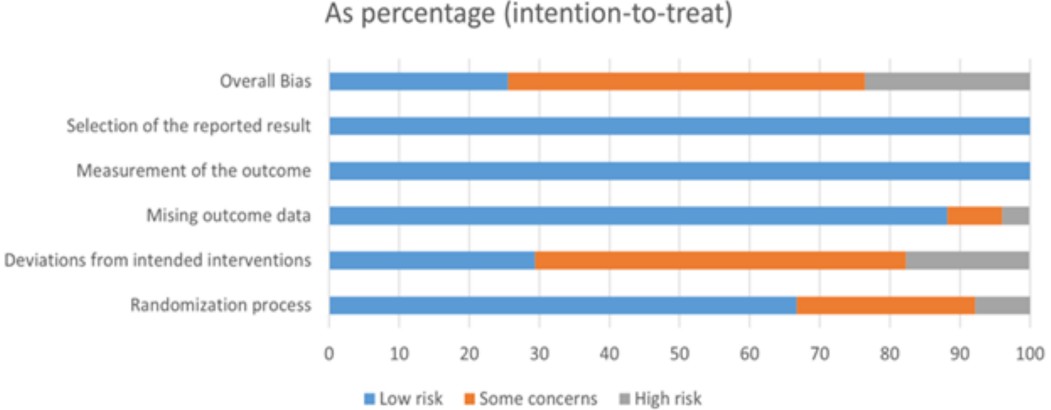

**Figure 2** Review authors' judgements about each risk of bias item presented as percentages across all included studies.

*2022*; *Xu & Fang, 2016*; *Yan, Zhao & Peng, 2022*; *Yang et al., 2021*; *Yao et al., 2020*; *Yu & Cao, 2018*; *Zeng et al., 2020*; *Zhou, 2020*; *Zhou et al., 2016*; *Zhu et al., 2020*) examined reported ALB. Comprehensive nursing care (MD = 6.01, 95% CrI [3.36–8.7]), resistance exercise (MD = 2.37, 95% CrI [0.2–4.55]), ONS (MD = 2.12, 95% CrI [0.51–3.75]), and dietary intervention (MD = 6.05, 95% CrI [4.32–7.76]), and health education (MD = 3.7, 95% CrI [1.14–6.26]) were superior to usual care in improving ALB, with statistically significant differences. Comprehensive nursing care and dietary intervention were statistically different from resistance exercise, aerobic exercise, ONS, aerobic exercise combined with resistance exercise, and ONS combined with aerobic exercise. No statistical difference was found between the other interventions ($P > 0.05$; Table 4).

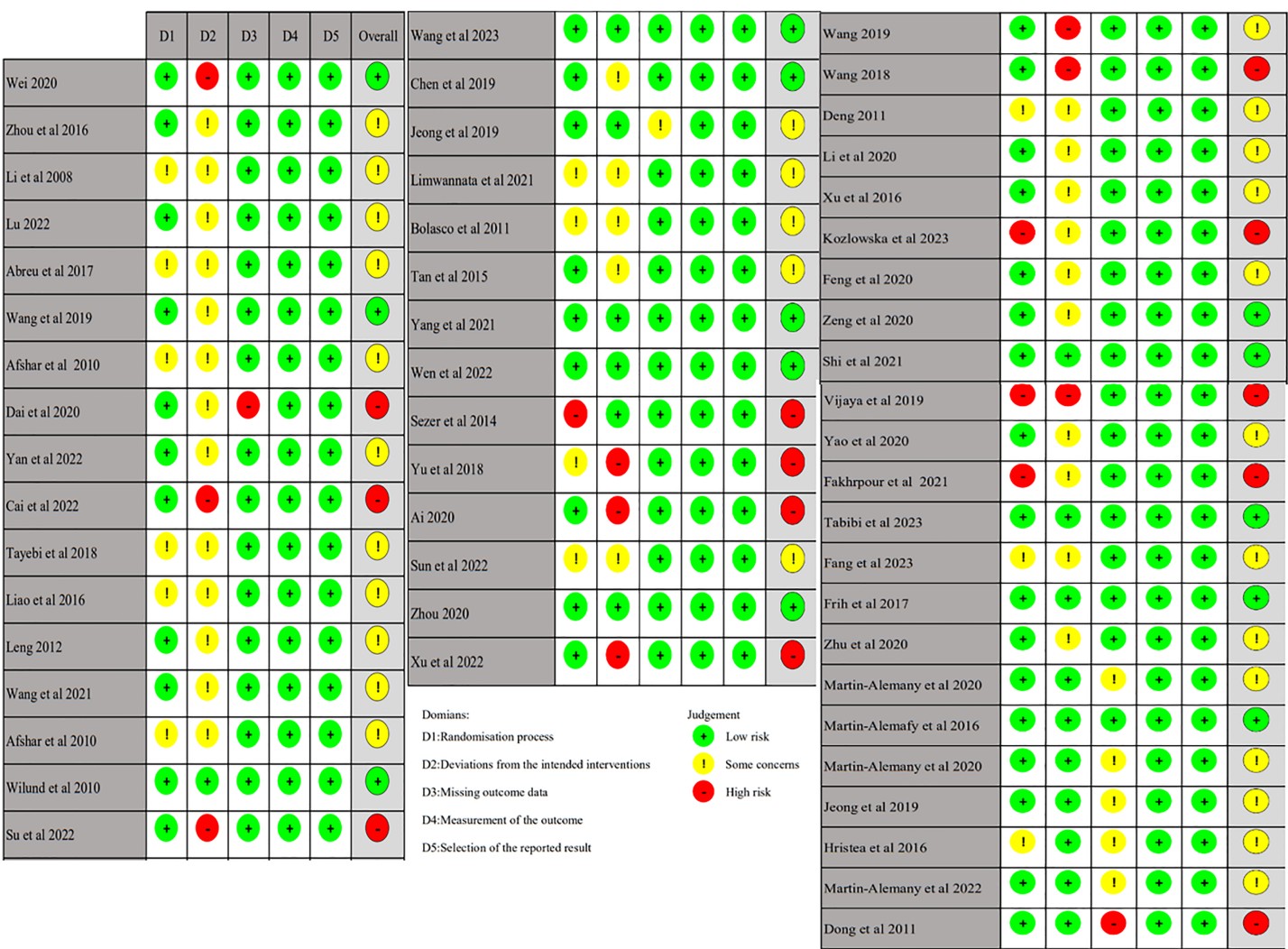

**Figure 3  Risk of bias summary** (*Abreu et al., 2017*; *Afshar et al., 2010*; *Ai, 2020*; *Bolasco et al., 2011*; *Cai et al., 2022*; *Chen, Zhao & Huang, 2019*; *Dai & Ma, 2021*; *Deng, 2011*; *Dong et al., 2011*; *Fakhrpour et al., 2020*; *Fang et al., 2023*; *Feng et al., 2020*; *Frih et al., 2017*; *Hristea et al., 2016*; *Jeong et al., 2019*; *Kozlowska et al., 2023*; *Leng, 2012*; *Li et al., 2008*; *Li & Feng, 2020*; *Liao et al., 2016*; *Limwannata et al., 2021*; *Lu, 2022*; *Martin-Alemañy et al., 2020, 2016, 2022*; *Sezer et al., 2014*; *Shi et al., 2021*; *Su et al., 2022*; *Sun, Sun & Yang, 2022a*; *Tabibi et al., 2023*; *Tan et al., 2015*; *Tayebi, Ramezani & Kashef, 2018*; *Vijaya et al., 2019*; *Wang & Liu, 2021*; *Wang, 2018*; *Wang et al., 2019*; *Wang, 2019*; *Wang et al., 2023*; *Wei, 2020*; *Wen et al., 2022*; *Wilund et al., 2010*; *Xu et al., 2022*; *Xu & Fang, 2016*; *Yan, Zhao & Peng, 2022*; *Yang et al., 2021*; *Yao et al., 2020*; *Yu & Cao, 2018*; *Zeng et al., 2020*; *Zhou, 2020*; *Zhou et al., 2016*; *Zhu et al., 2020*).

***Results of NMA on the effects of different interventions on HB in patients on MHD***

A total of 41 studies (*Abreu et al., 2017*; *Afshar et al., 2010*; *Ai, 2020*; *Bolasco et al., 2011*; *Cai et al., 2022*; *Chen, Zhao & Huang, 2019*; *Dai & Ma, 2021*; *Deng, 2011*; *Dong et al., 2011*; *Fakhrpour et al., 2020*; *Fang et al., 2023*; *Feng et al., 2020*; *Frih et al., 2017*; *Hristea et al., 2016*; *Kozlowska et al., 2023*; *Leng, 2012*; *Li et al., 2008*; *Lu, 2022*; *Martin-Alemañy et al., 2020*; *Sezer et al., 2014*; *Shi et al., 2021*; *Su et al., 2022*; *Sun, Sun & Yang, 2022a*; *Tabibi et al., 2023*; *Tan et al., 2015*; *Wang & Liu, 2021*; *Wang, 2018*; *Wang et al., 2019*; *Wang, 2019*; *Wang et al., 2023*; *Wei, 2020*; *Wen et al., 2022*; *Xu et al., 2022*; *Xu & Fang, 2016*; *Yan, Zhao*

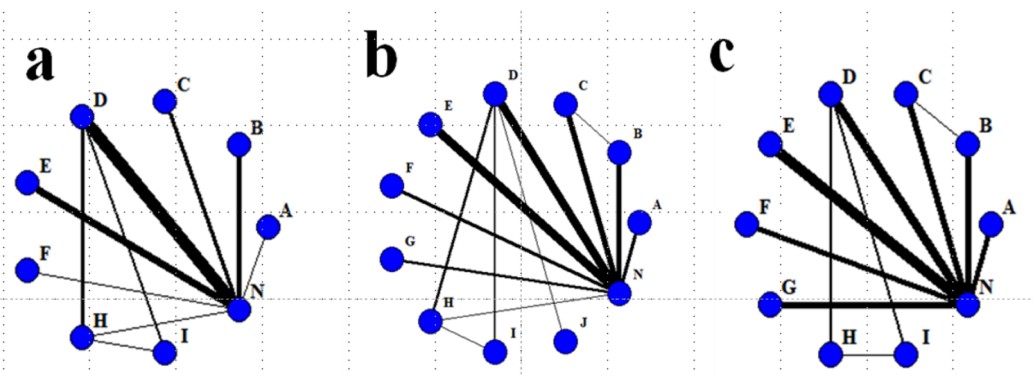

**Figure 4 Network evidence map for each outcome indicator.** (A) = BMI; (B) = ALB; (C) = HB A = Comprehensive nursing care; B = Resistance exercise; C = Aerobic exercise; D = ONS; E = Dietary intervention; F = Health education; G = Aerobic and resistance exercise; H = ONS combined with Aerobic exercise; I = ONS combined with Resistance exercise; J = ONS combined with Aerobic and Resistance exercise; N = Usual care. The circular nodes in the network diagrams represent the type of intervention, and the size of the nodes are related to the number of patients included in the corresponding intervention. The thickness of the line is related to the number of interventions compared with one another.



**Table 3 Results of NMA of the effects of different intervention phrases on BMI in patients on MHD.**

| Comprehensive nursing care | | | | | | | | |
|---|---|---|---|---|---|---|---|---|
| −0.91 (−4.1, 2.41) | Resistance exercise | | | | | | | |
| −0.65 (−4, 2.72) | 0.26 (−2.36, 2.8) | Aerobic exercise | | | | | | |
| −0.66 (−3.6, 2.26) | 0.25 (−1.81, 2.19) | −0.01 (−2.17, 2.14) | ONS | | | | | |
| −1.97 (−4.97, 1.11) | −1.06 (−3.23, 1.06) | −1.32 (−3.59, 1) | −1.31 (−2.89, 0.36) | Dietary intervention | | | | |
| −1.64 (−5.59, 2.33) | −0.72 (−4.11, 2.54) | −0.98 (−4.39, 2.42) | −0.98 (−3.96, 2.04) | 0.33 (−2.81, 3.41) | Health education | | | |
| −1.33 (−4.84, 2.15) | −0.43 (−3.23, 2.29) | −0.68 (−3.56, 2.21) | −0.67 (−2.64, 1.31) | 0.63 (−1.9, 3.12) | 0.3 (−3.26, 3.85) | ONS combined with Aerobic exercise | | |
| −1.99 (−5.64, 1.93) | −1.07 (−4.11, 2.11) | −1.34 (−4.44, 2) | −1.32 (−3.62, 1.23) | −0.02 (−2.83, 2.97) | −0.35 (−4.06, 3.63) | −0.65 (−3.18, 2.12) | ONS combined with Resistance exercise | |
| −0.16 (−2.92, 2.6) | 0.75 (−1.03, 2.44) | 0.49 (−1.41, 2.4) | 0.5 (−0.49, 1.5) | 1.81 (0.49, 3.06) | 1.48 (-1.35, 4.31) | 1.17 (−0.99, 3.32) | 1.83 (−0.9, 4.32) | Usual care |

*& Peng, 2022; Yang et al., 2021; Yao et al., 2020; Zeng et al., 2020; Zhou, 2020; Zhou et al., 2016; Zhu et al., 2020*) assessed changes in HB. Compared with usual care, comprehensive nursing care (MD = 9.09, 95% CrI [2.96–15.63]), aerobic exercise (MD = 5.36, 95% CrI [0.27–10.5]), ONS (MD = 5.03, 95% CrI [0.71–9.44]), dietary intervention (MD = 9.21,

**Table 4 Results of NMA of the effects of different intervention phrases on ALB in patients on MHD.**

| Comprehensive nursing care | | | | | | | | | | |
|---|---|---|---|---|---|---|---|---|---|---|
| 3.64 (0.2, 7.08) | Resistance exercise | | | | | | | | | |
| 4.72 (1.41, 8.05) | 1.08 (−1.75, 3.93) | Aerobic exercise | | | | | | | | |
| 3.9 (0.78, 7.01) | 0.25 (−2.46, 2.96) | −0.83 (−3.4, 1.74) | ONS | | | | | | | |
| −0.03 (−3.18, 3.17) | −3.68 (−6.45, −0.9) | −4.76 (−7.39, −2.12) | −3.93 (−6.27, −1.55) | Dietary intervention | | | | | | |
| 2.32 (−1.35, 6.03) | −1.33 (−4.68, 2.05) | −2.42 (−5.65, 0.87) | −1.58 (−4.59, 1.46) | 2.34 (−0.73, 5.43) | Health education | | | | | |
| 4.3 (0.36, 8.31) | 0.66 (−3.01, 4.35) | −0.42 (−3.99, 3.16) | 0.41 (−2.94, 3.79) | 4.34 (0.89, 7.76) | 1.99 (−1.93, 5.89) | Aerobic and resistance exercise | | | | |
| 5.93 (1.71, 10.16) | 2.28 (−1.64, 6.22) | 1.2 (−2.63, 5.04) | 2.02 (−1, 5.08) | 5.96 (2.26, 9.65) | 3.61 (−0.55, 7.77) | 1.62 (−2.77, 6.04) | ONS combined with Aerobic exercise | | | |
| 3.76 (−1.18, 8.72) | 0.12 (−4.58, 4.79) | −0.97 (−5.57, 3.62) | −0.13 (−4, 3.72) | 3.79 (−0.73, 8.28) | 1.44 (−3.45, 6.31) | −0.54 (−5.66, 4.56) | −2.16 (−6.67, 2.31) | ONS combined with Resistance exercise | | |
| 0.6 (−15.36, 16.61) | −3.04 (−18.94, 12.86) | −4.14 (−19.99, 11.8) | −3.3 (−18.94, 12.42) | 0.62 (−15.18, 16.5) | −1.72 (−17.66, 14.3) | −3.7 (−19.66, 12.39) | −5.34 (−21.23, 10.66) | −3.16 (−19.24, 12.97) | ONS combined with Aerobic and Resistance exercise | |
| 6.01 (3.36, 8.7) | 2.37 (0.2, 4.55) | 1.29 (−0.7, 3.29) | 2.12 (0.51, 3.75) | 6.05 (4.32, 7.76) | 3.7 (1.14, 6.26) | 1.72 (−1.25, 4.68) | 0.1 (−3.19, 3.36) | 2.26 (−1.89, 6.42) | 5.42 (−10.36, 21.13) | Usual care |

95% CrI [5.05–13.42]), health education (MD = 10.76, 95% CrI [5.43–16.01]), and aerobic exercise combined with resistance exercise (MD = 6.57, 95% CrI [1.16–12.02]) were more effective in improving the patients' HB levels. Significant difference was found between resistance exercise and health education, whereas no significant differences were found among the other interventions ($P > 0.05$; Table 5).

## Transitivity and inconsistency analysis

A PRSF of 1 for each outcome indicator suggests that the convergence of a model is good and that the model constructed in this study can effectively reflect relevant data. The convergence diagnosis and trajectory density plots of the model are shown in Appendices C and D.

The results of the node-splitting method showed that the 95% CrI of each outcome indicator was 0 in the node-splitting method and at $P$ of >0.05, indicating that the results of the indirect comparison were consistent with those of the direct comparison. No statistical inconsistency was found, indicating good consistency among the closed loops (Appendix E).

**Table 5 Results of NMA of the effects of different intervention phrases on HB in patients on MHD.**

| Comprehensive nursing care | | | | | | | | | |
|---|---|---|---|---|---|---|---|---|---|
| 5.66 (−2.18, 13.97) | Resistance exercise | | | | | | | | |
| 3.73 (−4.2, 12.06) | −1.92 (−8.62, 4.61) | Aerobic exercise | | | | | | | |
| 4.06 (−3.48, 11.91) | −1.6 (−8.33, 5.01) | 0.34 (−6.42, 7.03) | ONS | | | | | | |
| −0.12 (−7.54, 7.64) | −5.77 (−-12.36, 0.68) | −3.85 (−10.47, 2.75) | −4.18 (−10.22, 1.87) | Dietary intervention | | | | | |
| −1.68 (−9.69, 6.85) | −7.31 (−14.61, −0.02) | −5.4 (−12.73, 2.05) | −5.72 (−12.52, 1.23) | −1.55 (−8.23, 5.27) | Health education | | | | |
| 2.5 (−5.65, 11.04) | −3.14 (−10.56, 4.2) | −1.21 (−8.64, 6.27) | −1.54 (−8.46, 5.47) | 2.62 (−4.22, 9.53) | 4.17 (−3.46, 11.74) | Aerobic and resistance exercise | | | |
| 6.87 (−4.31, 19.04) | 1.23 (−9.46, 12.41) | 3.14 (−7.51, 14.45) | 2.81 (−5.52, 11.92) | 6.99 (−3.25, 17.96) | 8.54 (−2.25, 19.9) | 4.36 (−6.47, 15.8) | ONS combined with Aerobic exercise | | |
| 0.15 (−14.19, 14.77) | −5.54 (−19.53, 8.35) | −3.6 (−17.5, 10.35) | −3.94 (−16.16, 8.32) | 0.24 (−13.39, 13.92) | 1.78 (−12.23, 15.83) | −2.38 (−16.46, 11.66) | −6.78 (−20.78, 6.86) | ONS combined with Resistance exercise | |
| 9.09 (2.96, 15.63) | 3.44 (−1.59, 8.4) | 5.36 (0.27, 10.5) | 5.03 (0.71, 9.44) | 9.21 (5.05, 13.42) | 10.76 (5.43, 16.01) | 6.57 (1.16, 12.02) | 2.22 (−7.85, 11.59) | 8.97 (−4.05, 21.97) | Usual care |

## Model selection

After preliminary model fitting, the DIC values under the random- and fixed-effects models for each outcome indicator (BMI, ALB, and HB) were small, and difference was greater than 5. The $I^2$ value for each outcome indicator (in the random-effects model, the value was less than 50% (BMI: $I^2$ = 3%, ALB: $I^2$ = 2, and HB: $I^2$ = 3%), and thus the random-effects and consistency models were analyzed for all the present studies (Appendix F).

## Rank probabilities

The SUCRA results indicated that dietary intervention was most effective in improving BMI and ALB in MHD patients, while health education showed the greatest effect on HB. The rankings and cumulative ranked probability plots for each of the remaining interventions for the different outcome indicators are shown in Appendices G and H.

## Publication bias

Funnel plots and Egger's tests showed no significant publication bias in any studies included for each outcome metric (Appendix I).

## Sensitivity analysis

In the sensitivity analysis, we eliminated all the studies one by one to find the source of heterogeneity. In ALB and HB, the serum biochemical indicators introduced by *Cai et al.*

*(2022)* were measured before the first dialysis after the start of the study. By contrast, they were measured before the start of the study in the other studies and may have thus been the source of heterogeneity. The heterogeneity in the rest of the studies originated from differences in the number of control and intervention groups. Notably, the results of meta-analysis of resistance exercise, aerobic exercise combined with resistance exercise, comprehensive nursing care, and ONS on HB were unstable. Therefore, caution was needed in the analysis of the above results. Nevertheless, the rest of the results were robust (Appendix J).

### Results of the GRADE quality of evidence evaluation

The quality of evidence on the improvement of BMI, ALB, and HB in MHD patients with different nonpharmacological interventions was assessed with the GRADE system, which scores the quality of direct, indirect, and network effect estimates according to the NMA model proposed by *Puhan et al. (2014)*. In terms of improving ALB, two pairs of interventions had a moderate level. For HB improvement, two pairs of interventions had a moderate level and the rest were all rated as low or very low levels of evidence. Most of the studies were downgraded by inaccuracy, risk of bias, and inconsistency (Appendix K).

## DISCUSSION

This study is the first Bayesian NMA of the effects of 11 nonpharmacological interventions on nutrition-related indicators (BMI, ALB, and HB) in patients on MHD, and 54 studies were included. The results suggested that among all interventions, dietary intervention is the best intervention program for BMI and ALB in MHD patients and health education is the most effective in improving HB in patients. ONS combined with resistance exercise and health education ranked second and third, respectively, in terms of improving patients' BMI. As for the ALB levels of the patients, comprehensive nursing care and health education ranked second and third, respectively. In terms of improving HB, dietary intervention and comprehensive nursing care ranked second and third, respectively.

### Dietary intervention is the most effective in improving BMI and ALB in patients on MHD

BMI is an important indicator for evaluating the prognoses of dialysis patients and whether a patient has a normal weight (*Suzuki et al., 2020*). It is an important factor for predicting whether a patient has pulmonary infection and cardiovascular disease. Notably, it affects the readmission rate of patients on MHD (*Tang, 2019*). Previous systematic reviews and meta-analyses (*Herselman et al., 2010*; *Jialin, Yi & Weijie, 2012*) have shown that BMI has a significant negative correlation with all-cause mortality rate in dialysis patients, and this relationship is prevalent in retrospective studies and large-sample studies. which is consistent with the long-term follow-up results of a large number of dialysis patients conducted by *Kim et al. (2020)* and was reconfirmed in a recent review (*Rabbani et al., 2022*).

Inadequate food intake due to loss of appetite and dietary restriction may predispose patients to malnutrition, and malnourished HD patients have low BMI and high mortality

rates (*de Mutsert et al., 2009*). By contrast, a healthy diet reduces age-adjusted all-cause mortality rate in patients with CKD (*Rysz et al., 2017*). A personalized dietary intervention ensures that patients consume sufficient nutrients and prevents the excessive intake of unhealthy foods, improving appetite and digestive function, compensating for the negative energy balance due to the loss of nutrients during dialysis, and increasing in patients' weight and BMI (*Ai, 2020*; *Deng, 2011*). The ranking of SUCRA values in this study suggests that dietary intervention is the most effective intervention for improving BMI in MHD patients. A systematic evaluation and meta-analysis has shown that the effect of dietary intervention on BMI is uncertain (*Palmer et al., 2017*), but our study and other studies have shown that dietary intervention improves the BMI of patients (*Ai, 2020*; *Li & Feng, 2020*; *Wang, 2019*; *Xu & Fang, 2016*). This finding may be related to the fact that our study was more focused on the MHD group or the low quality of the studies included. Therefore, whether dietary intervention improves patients' BMI should be validated, and high-quality RCTs are needed to validate the reliability of dietary intervention on BMI outcomes in patients on MHD.

ALB is a nutrient and associated with capillary pressure in the body and one of the most commonly used laboratory indicators for evaluating nutritional status (*Chen, 2020*). A decrease in ALB level indicates liver and kidney dysfunction and impaired absorption of nutrients (*Yan, Zhao & Peng, 2022*). A decrease in a patient's appetite affects the body's intake of protein and energy, leading to malnutrition and a considerable increase in the risk of mortality (*Shen, Huang & Han, 2023*). In this study, SUCRA ranking shows that dietary intervention can effectively improve the ALB levels of patients. This result is consistent with the findings of a systematic review (*Palmer et al., 2017*) and other scholars (*Kozlowska et al., 2023*; *Li & Feng, 2020*; *Sun, Sun & Yang, 2022a*; *Viramontes Hörner et al., 2020*; *Wang, 2018*; *Yamashita et al., 2018*). Dietary intervention emphasized the intake of high-quality protein (*Ai, 2020*; *Deng, 2011*), First, high-quality protein helps synthesize and maintain plasma ALB levels; Secondly, it can promote the absorption and utilization of nutrients and provide more raw materials for the synthesis of plasma ALB, thereby improving the level of ALB. Finally, a stable level of plasma ALB can be effectively maintained by increasing protein intake and reducing unnecessary protein loss.

In summary, dietary intervention improves patients' nutritional and biochemical indicators. Regulating the dietary structure of patients and formulating a personalized and precise dietary management plan not only helps to increase the protein intake of patients, reduce nutrient loss due to insufficient dialysis, and improve the nutritional status of patients but also improves their quality of life and survival rates (*Li & Feng, 2020*). In addition, dietary intervention can increase patients' knowledge and awareness of dialysis-related dietary knowledge and reducing the occurrence of related complications due to lack of knowledge. However, ordinary dietary guidance does not greatly improve the nutritional status of patients (*Sun, Sun & Yang, 2022a*). Therefore, targeted dietary care interventions need to be adopted in clinics to guide patients and help them to adopt a reasonable diet to improve their nutritional status.

## Health education is the most effective in improving HB in patients on MHD

HB in MHD patients is an important indicator for reflecting the long-term quality of life and prognoses of patients (*Li et al., 2022*), and the normal range is between 110 and 130 g/L (*Chen, Sun & Cai, 2021*). However, the compliance rate for HB in MHD patients in China is only 20% (*Locatelli et al., 2004*) possibly because of the lack of relevant knowledge and delayed treatment of patients. This situation may also be related to insufficient nutrition intake due to loss of appetite (*Liu, Liu & Ping, 2023b*). Decrease in HB level not only affects the quality of life of patients but also increases the risk of infection (*Dalrymple et al., 2010*; *Sun, Yang & Zhou, 2020*). The cost of anemia treatment is considerably high (*Spinowitz et al., 2019*), and even though current treatment protocols can alleviate this situation, a patient's financial burden remains high. Thus, economical and effective treatment options for dialysis patients are needed. The SUCRA ranking of this study indicates that health education can effectively improve HB levels in MHD patients, consistent with the results of several studies (*Feng et al., 2020*; *Yao et al., 2020*; *Zeng et al., 2020*). The "Internet + Health education" mode breaks the boundaries of time and space and provides convenient and personalized health guidance for MHD patients *via* network platforms, such as WeChat, QQ, mini programs, and hospital cloud follow-up (*Shi et al., 2021*). In this mode, patients and their families can obtain individualized diet and health education programs, including diet adjustment and disease cognition improvement. Dietary guidance is particularly critical (*Liu, Zhao & Liu, 2023c*), which emphasizes high-quality protein and iron intake. These measures not only enhance the treatment compliance of patients but also promote the synthesis and stability of HB, improve treatment effects, reduce the occurrence of complications, and improve the quality of life of patients (*Ai, 2020*; *Feng et al., 2020*; *Zeng et al., 2020*). Compared with drug therapy, health education is a more economical treatment plan. However, health education does not result in statistically significant difference in HB level (*Shi et al., 2021*) possibly because of short intervention period. Hence, improvement in HB is not evident. Therefore, large and rigorous RCTs with extended intervention periods must be adopted in clinics to verify the positive effect of health education on the HB levels of MHD patients and facilitate the development of a reasonable health education program that can enhance health and nutritional status.

## Implications for future research and clinical practice

Our results show that nonpharmacological interventions can effectively improve the nutritional status of patients and can be applied to clinical practice. In future clinical care, we need to consider individual factors, "prescribe the right medicine," and select appropriate interventions according to patients' clinical characteristics and biochemical index results. In addition, joint personalized intervention programs based on different populations and cultural backgrounds should be developed to guide patients and encourage them to have a reasonable diet. The mode of intervention (*e.g.*, offline or a combination of offline and online), type of intervention (videos and lectures), time (during dialysis or post-dialysis), environment of the implementation, barriers, facilitators, and the

degree of acceptance by the patient should be considered in implementing interventions. This approach can increase patients' recognition degree regarding the implementation of interventions. In addition, we should clarify the therapeutic mechanisms of different nonpharmacological interventions and why some nonpharmacological interventions are more effective than others in improving some nutritional indicators, which will be more effective in providing help to MHD patients in improving their nutritional status (*Sun et al., 2022b*). However, the correlative evidence in this study is weak, and further exploration is needed to examine the effects of these nonpharmacological interventions in different healthcare settings in future studies, which will ensure the reliability and validity of interventions (*Wei et al., 2022*). Therefore, well-designed, high-quality, and double-blind RCTs are needed to further validate currently available evidence.

### Strengths and limitations

**Strengths:** This study is the first to compare the effects of nonpharmacological interventions on the nutritional status of patients on MHD through Bayesian NMA, considerably improving the credibility of the results. We conducted a comprehensive and systematic search in eight commonly used databases to evaluate all nonpharmacological interventions that affect the nutritional status of patients on MHD and finalize our outcome indicators. We used SUCRA to rank the efficacy of different interventions to identify the best intervention and used GRADE to assess the quality of evidence. Then, we performed sensitivity analyses on NMA to ensure the robustness of our results.

**Limitations:** The literature included in this study was highly heterogeneous, and the sample sizes of most of the included studies were small, possibly affecting the applicability and accuracy of the results (*Sun et al., 2022b*). The experimental designs of the studies were not rigorous, and most studies were randomized and did not blind the investigators and experimenters. Thus, most studies presented outcomes considered "high risk" or "some concern." We only analyzed the effects of the interventions on BMI and HB and ALB levels, and other anthropometric and biochemical indicators were not included. Thus, there are certain limitations in evaluating the improvement of nutritional status of MHD patients by nonpharmacological interventions only based on the above three indicators. In addition, we did not conduct subgroup analyses on baseline characteristics, and most studies had short intervention cycles. Whether country, male-to-female ratio, gender, age, and intervention period affected our results and whether the effects of these interventions vary depending on the length of the intervention cycle remain unclear. The quality of our evidence was rated as very low or low, and therefore our results are not conclusive evidence. Additional studies and rigorous trial designs are needed to validate these results.

## CONCLUSIONS

This study is the first NMA of the effectiveness of nonpharmacological interventions in improving nutrition-related indicators in MHD patients. The results of the analysis showed that dietary interventions performed better in improving patients' BMI and ALB levels than other interventions and usual care, suggesting that by adjusting patients' diets is effective in improving their nutritional status and improves treatment outcomes and

quality of life. Our analysis also found that health education improves patients' HB levels possibly because it can help patients to better understand and manage their diseases and improve their treatment compliance, thus contributing to improvement in nutrition-related problems, such as anemia. The results of these NMA provide new perspectives and insights for clinical caregivers, who can develop more personalized intervention plans according to the specific needs and conditions of patients, that is, focusing on strengthening dietary interventions for patients with low BMI and ALB levels and enhancing health education to improve self-management for patients with low HB levels. However, we the limitations of this study should be addressed. Owing to the low quality of evidence from the included studies, our conclusions may have some shortcomings. Therefore, in future clinical trials, we need to include high-quality and large RCTs to further validate the therapeutic effects of nonpharmacological interventions on the nutritional status of patients with MHD and accurately assess the validity and reliability of these interventions for the identification of the most precise and effective treatment.

# ACKNOWLEDGEMENTS

The authors thank all the reviewers for their assistance and support.

## Funding

This work was supported by Natural Science Foundation of Xinjiang Uygur Autonomous Region (2018D01C196). The funders had no role in study design, data collection and analysis, decision to publish, or preparation of the manuscript.

## Grant Disclosures

The following grant information was disclosed by the authors:
Natural Science Foundation of Xinjiang Uygur Autonomous Region: 2018D01C196.

## Competing Interests

The authors declare that they have no competing interests.

## Author Contributions

- Xiaolan Ma conceived and designed the experiments, performed the experiments, analyzed the data, prepared figures and/or tables, and approved the final draft.
- Chun Tang performed the experiments, authored or reviewed drafts of the article, and approved the final draft.
- Hong Tan performed the experiments, analyzed the data, prepared figures and/or tables, and approved the final draft.
- Jingmei Lei performed the experiments, analyzed the data, prepared figures and/or tables, and approved the final draft.
- Li Li conceived and designed the experiments, analyzed the data, authored or reviewed drafts of the article, and approved the final draft.

## Data Availability

This is a systematic review/meta-analysis.

## Supplemental Information

Supplemental information for this article can be found online at http://dx.doi.org/10.7717/peerj.19053#supplemental-information.

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
