# Peer review of "Comparative effectiveness of nonpharmacological interventions for the nutritional status of maintenance hemodialysis patients: a systematic review and network meta-analysis"

_PeerJ, doi:10.7717/peerj.19053_

## Round 0.1 · original submission · Major Revisions

The manuscript is very difficult to read because of the extensive number of abbreviations, many of them unusual - so, please write more often the entire expression instead of abbreviation. Also, what the dietary interventions were should be clearly explained. In addition, please pay careful attention (and adequately resolve) the comments from reviewers 2 and 4.

·

Basic reporting

no comment

Experimental design

no comment

Validity of the findings

no comment

Additional comments

There is a lot of abbreviation
Some statement are not clear, like what is the meaning of health education , diet therapy , oral nutritional therapy? , these statement need definition and are these interventions are fixed with all references ??
the scientific content is little

·

Basic reporting

The manuscript presents a high-quality meta-analysis on the efficacy of nonpharmacological interventions in the improvement of nutritional status in patients undergoing hemodialysis (HD). The English language used is generally good, with only minor areas for improvement. The figures and tables are relevant and well-labeled. The references are comprehensive and also relevant.
The introduction provides adequate context, highlighting the importance of nonpharmacological interventions for the nutritional status of HD patients. However, the research gap question in knowledge this study aims to fill.

Experimental design

The study is original and falls within the scope of the journal. The research question is well-defined. The authors have appropriately followed PRISMA guidelines, and the study is registered on PROSPERO. It meets ethical standards, and the methodology is sound. However, some additional information on the intervention examined is required (please see the additional comments).

Validity of the findings

The data used in the analysis are robust, with a sufficient number of studies (54) and patients (3861) included. The results are statistically sound and controlled, and publication bias is adequately addressed. The conclusions are aligned with the original research question but could be rephrased to ensure clarity.

Additional comments

The authors have done great work and are to be commended for their thorough approach to this important clinical question. I have no significant comments. My suggestions are as follows:
1) Minor English language polishing is needed. The paper is overloaded with abbreviations; please consider reducing their use.
2) The abstract could be more concise and structured.
3) The authors concluded that dietary intervention is more effective than other examined non-pharmacological interventions. However, they do not elaborate on what exactly the dietary interventions entailed across the included studies. So, the best diet intervention is unclear. The authors should add more information on the types of dietary interventions that were included (e.g. low protein, Mediterranean diet, etc.).
4) In the Discussion, potential mechanisms for why certain interventions may be most effective should be provided.
5) Please rephrase the conclusions to ensure that the main findings are communicated clearly.

Reviewer 3 ·

Basic reporting

After observing the manuscript I found
1- No Plagiarism
2- References between up-to-date and recent
3- There is no conflict of interest between the authors (Xiaolan Ma, Chun Tang, Hong Tan, Jingmei Lei and Li Li study conception and design, data collection, data analysis, interpretation, drafting of the manuscript, and critical revision of the manuscript. All authors contributed to the article and approved the submitted version)

Experimental design

Academically written according to correct scientific writing

Validity of the findings

The results are suitable for research and accurate and Validity

Additional comments

I have no comments

Reviewer 4 ·

Basic reporting

I believe that the Network meta-analysis is rich and relevant, with significant practical guidance and policy implications. Further, by and large, I find the paper to have used robust methodology, specifically for all the steps used to reduce bias. This makes the manuscript a good candidate for a possible publication.

However, I believe that, as it currently stands, the manuscript is not ready for publication and needs to be improved. In the section below I lay out a number of comments and suggestions that are likely to improve the manuscript and that the authors might wish to take into account.

Following this journal review guidance:
- English proficiency should be improved
- Raw data is shared and comprehensive
- I have some general comments on the structure and figures used

Experimental design

The research used a rigorous methodology, explained in details across the manuscript and the Appendices provided

Validity of the findings

Very valid, yet very hard to read and follow

Additional comments

• The result sections are ill-structured and ill-written
o Spell all phrases before abbreviations (in text too): RE?
o The section of “characteristics of the included studies” is very confusing to read- mentioning all included studies in unnecessary, especially where the table presented all.
o Figure 1: there is a mis-match between the numbers included in figure 1 and in text under manuscript section 3.1
o Re-write section 3.4: maybe add ID numbers for each study? with adding sub-sections for each outcome?

• General comments:
o Triceps skinfold thickness was mentioned in the introduction as one of the major anthropometric effectiveness indicators, is there an explanation why it wasn’t considered as a tested outcome?

---

## Round 0.2 · Minor Revisions

Please check the references carefully and try to improve the understandability of the manuscript.

Reviewer 4 ·

Basic reporting

- Great improvements from the first submitted manuscript, still think some English editing would be useful

- Most provided comments from the first review round were considered

- Still think some parts are "too rich and hard to read", specially in the results section- figures/ tables should make it easier

Experimental design

I still think the methodology is solid and suitable for the research question

Validity of the findings

-

Additional comments

- Error in the citation tool used. the first referee is " !!! INVALID CITATION !!! . " and coming up in the manuscript multiple times

---

## Round 0.3 · accepted · Accept

No further comments on this revised version.

Reviewer 4 ·

Basic reporting

No needed changes

Experimental design

-

Validity of the findings

-

Additional comments

-